# Middle-Out Decoding

**Shikib Mehri**
Department of Computer Science
University of British Columbia
amehri@cs.cmu.edu

**Leonid Sigal**
Department of Computer Science
University of British Columbia
lsigal@cs.ubc.ca

## Abstract

Despite being virtually ubiquitous, sequence-to-sequence models are challenged by their lack of diversity and inability to be externally controlled. In this paper, we speculate that a fundamental shortcoming of sequence generation models is that the decoding is done strictly from left-to-right, meaning that outputs values generated earlier have a profound effect on those generated later. To address this issue, we propose a novel *middle-out decoder* architecture that begins from an initial middle-word and simultaneously expands the sequence in both directions. To facilitate information flow and maintain consistent decoding, we introduce a *dual self-attention* mechanism that allows us to model complex dependencies between the outputs. We illustrate the performance of our model on the task of video captioning, as well as a synthetic sequence de-noising task. Our middle-out decoder achieves significant improvements on de-noising and competitive performance in the task of video captioning, while quantifiably improving the caption diversity. Furthermore, we perform a qualitative analysis that demonstrates our ability to effectively control the generation process of our decoder.

## 1 Introduction

Neural encoder-decoder architectures have gained significant popularity in tasks such as language translation (Sutskever et al., 2014; Bahdanau et al., 2015; Wu et al., 2016), image captioning (Vinyals et al., 2015; Xu et al., 2015), video captioning (Venugopalan et al., 2015a), visual question answering (Antol et al., 2015; Malinowski et al., 2015) and many others. In such architectures, the input is typically encoded through the use of an *encoder* (*e.g.*, a CNN for an image or RNN, with LSTM or GRU cells, for a sentence or a variable length input) into a fixed-length vector hidden representation. The *decoder*, for most problems, is then tasked with generating a sequence of words, or, more generally, vector values, conditioned on the hidden representation of the encoder.

One issue with such encoder-decoder architectures is that conditioning is done through the initial hidden state of the decoder (Vinyals et al., 2015) and has to be propagated forward to generate a relevant output sequence. This design makes it difficult, for example, to generate relevant image captions as the encoded image information is often overwhelmed by the language prior implicitly learned by the decoder (Devlin et al., 2015; Heuer et al., 2016). Among potential solutions is the use of an encoded feature vector as a recurrent input to *each* decoder cell (Mao et al., 2015) or the use of attention over the encoder (Xu et al., 2015) to focus generation of each subsequent output. Modeling of long dependencies among the outputs in the decoded sequence is another challenge, often addressed by self-attention (Werlen et al., 2018; Daniluk et al., 2017; Vaswani et al., 2017).

In the existing architectures, however, the decoding is done strictly from left-to-right meaning that earlier generated output values have a profound effect on the later generated tokens. This makes the decoding (linguistically) consistent, but at the same time limited in diversity. Left-to-right decoding also presents challenges when one wants to have some control over a word or value that should appear later in the sequence. Ensuring that a particular important word explicitly appears in the caption is not possible with any of the left-to-right decoding mechanisms. Similarly, for decoding of symmetric sequences, a left-to-right decoding may introduce biases which break natural symmetries.

In this paper, we propose a novel *middle-out decoder* architecture that addresses these challenges. The proposed architecture consists of two RNN decoders, implemented using LSTMs, one which decodes to the right and another to the left; both begin from the same important (middle) output word or value automatically predicted by a classifier. Output can also be directly controlled by specifying an important (middle) word or value (*e.g.*, allowing controlled and more diverse caption generation focused on a specific verb, adjective, or object). As we show in experiments such ability is difficult to achieve using traditional left-to-right decoder architectures. During training and inference of our middle-out decoder the left and right LSTMs alternate between generating words/values in the two directions until they reach their respective STOP tokens. Further, a novel dual self-attention mechanism, that attends over both the generated output and hidden states, is used to ensure information flow and consistent decoding in both directions. The proposed architecture is a generalization of a family of traditional decoders that generate sequences left-to-right, where the important (middle) output is deterministically taken to be a sentence START token. As such, our middle-out decoder can be used in any modern architecture that requires sequential decoding.

**Contributions:** Our main contribution is the *middle-out decoder* architecture, which is able to decode a sequence starting from an important word or value in both directions. In order to ensure the middle-out generation is consistent, we also introduce a novel *dual self-attention* mechanism. We illustrate the effectiveness of our model through qualitative and quantitative experiments where we show that it can achieve state-of-the-art results. Further, we show vastly improved performance in cases where an important word or value can be predicted well (*e.g.*, decoding of sequences that have symmetries, or video captioning where an action verb can be estimated reliably).

## 2 Related Work

**Neural Encoder-Decoder Architectures:** Early neural architectures for language generation (Sutskever et al., 2014) produced sequential outputs, conditioned on a fixed-length hidden vector representation. The introduction of the attention mechanism (Bahdanau et al., 2015), allowed the language decoder to attend to the encoded input rather than necessitating that it be compressed into a fixed-length vector. Similar encoder-decoder architectures, with CNN encoding (Vinyals et al., 2015) for images and CNN+RNN encoding (Venugopalan et al., 2015a) for videos, were employed for the tasks of captioning and visual question answering (Antol et al., 2015; Malinowski et al., 2015). The proposed middle-out decoding focuses specifically on the decoder component of such models.

**Self-attention:** Recently, there have been a number of studies exploring the use of self-attention to improve language decoding (Werlen et al., 2018; Daniluk et al., 2017; Cheng et al., 2016; Liu and Lapata, 2017; Vaswani et al., 2017). Werlen et al. (2018) attend over the embedded outputs of the decoder, allowing for non-sequential dependencies between outputs. Daniluk et al. (2017) propose the self-attentive RNN, which attends over its own past hidden states. Vaswani et al. (2017) present the Transformer, an architecture that produces language, without a sequential model (*e.g.* an LSTM) and instead relies entirely on a number of attention mechanisms, including self-attention in the decoder. We utilize a novel combination of these self-attention mechanisms in our generation, which we refer to as dual self-attention.

**Non-traditional Decoders:** Most decoder architectures are sequential and generate sequences left-to-right (Sutskever et al., 2014), which leads to challenges such as lack of control and diversity (Devlin et al., 2015). Very recently tree structured decoders (Polosukhin and Skidanov, 2018) have been employed for tasks where outputs form well-defined structured representations (*e.g.*, programs). Our model makes no assumptions about the structure of the output sequence, beyond the notion that a certain element of the sequence may be more important than others.

**Controlled Decoding:** Hokamp and Liu (2017) introduced Grid Beam Search, an algorithm that enables controlled language generation, which they employ for the task of machine translation. Their method makes no architectural changes, and instead modifies beam search decoding to force the inclusion of certain phrases. Post and Vilar (2018) introduces a faster algorithm with the same effects. While our controlled generation is not as flexible as Grid Beam Search (Hokamp and Liu, 2017), we demonstrate that our model outperforms it in a confined, albeit realistic, setting.

**Diversified Decoding:** There has been some exploration into diverse language generation, particularly for the task of image captioning. Dai et al. (2017) uses a Conditional GAN, along with a policy

gradient, to generate diverse captions. Wang et al. (2016) presents GroupTalk, a framework which simultaneously learns multiple image caption distributions, to mimic human diversity.

**Video Captioning:** While our models are not specific to any particular task, we illustrate their performance on video captioning (Chen and Dolan, 2011). As such, it is related to work done in this problem space (Venugopalan et al., 2015b,a; Yao et al., 2015; Pan et al., 2016a,b; Yu et al., 2016; Song et al., 2017; Pasunuru and Bansal, 2017; Yu et al., 2017) to which we compare our results.

## 3 Approach

We now introduce our middle-out decoder, a novel architecture for the generation of sequential outputs from the middle outwards. Unlike standard decoders which generate sequences from left-to-right, our middle-out decoder simultaneously expands the sequence in both directions. Intuitively, doing so allows us to begin decoding from the most important component of a sequence, thereby allowing greater control over the resulting sequence generation.

We begin by discussing a baseline attention sequence-to-sequence model (Bahdanau et al., 2015). Next, we discuss the self-attention mechanism (Daniluk et al., 2017; Werlen et al., 2018), and present our novel *dual self-attention* applied to our baseline model. We then introduce our *middle-out decoder*, along with a classification model to predict the initial middle word.

### 3.1 Baseline Attention Sequence-to-Sequence Model

Our baseline model architecture, visualized in Figure 1, is that of Bahdanau et al. (2015): a bidirectional LSTM encoder and a unidirectional LSTM decoder. The input sequence is first passed through the encoder to produce a hidden state, $\mathbf{h}_i^e$, for each time-step $i$ of the input sequence. To produce the output sequence, we compute the hidden state at time $t$ as a function of the previous decoder hidden state $\mathbf{h}_{t-1}^d$, the embedding of the previously generated word $\mathbf{e}_{t-1}$ and the context vector $\mathbf{c}_t$:

$$\mathbf{h}_t^d = S(\mathbf{h}_{t-1}^d, \mathbf{e}_{t-1}, \mathbf{c}_t). \tag{1}$$

The context vector $\mathbf{c}_t$, is a weighted sum over the encoder hidden states:

$$\mathbf{c}_t = \sum_{i=1}^n \boldsymbol{\alpha}_{t,i} \mathbf{h}_i^e. \tag{2}$$

The alignment weights, $\boldsymbol{\alpha}_{t,i}$, allow our model to attend to different time-steps of the encoder dependant on the current hidden state of the decoder. We compute the weights as follows:

$$\boldsymbol{\alpha}_{t,i} = \frac{exp(score(\mathbf{h}_{t-1}^d, \mathbf{h}_i^e))}{\sum_{k=1}^n exp(score(\mathbf{h}_{t-1}^d, \mathbf{h}_k^e))}. \tag{3}$$

The score function used is the *general* attention mechanism described in Luong et al. (2015):

$$score(\mathbf{h}_{t-1}^d, \mathbf{h}_i^e) = \mathbf{h}_{t-1}^d \mathbf{W}_a \mathbf{h}_i^e. \tag{4}$$

Upon computing $\mathbf{h}_t^d$, we determine the output distribution, $p(\mathbf{w}_t|\mathbf{h}_t^d)$ through a softmax over all of the words in the vocabulary.

### 3.2 Self-Attention Mechanism

In our baseline architecture, non-adjacent time-steps of the decoder may only communicate through the propagation of a fixed-length vector, namely the hidden state of the LSTM. This hinders the model from learning long-term dependencies, which leads to a degraded quality for long output sequences. Bahdanau et al. (2015) observed a similar issue on the encoder side and subsequently introduced the attention mechanism. Recent work (Daniluk et al., 2017; Werlen et al., 2018; Vaswani et al., 2017) has proposed a self-attention mechanism to better capture the context of the decoder.

Werlen et al. (2018) attend over the embedded outputs of the decoder, thereby modifying the recurrence function to be:

$$\mathbf{h}_t^d = S(\mathbf{h}_{t-1}^d, \mathbf{d}_t, \mathbf{c}_t) \tag{5}$$

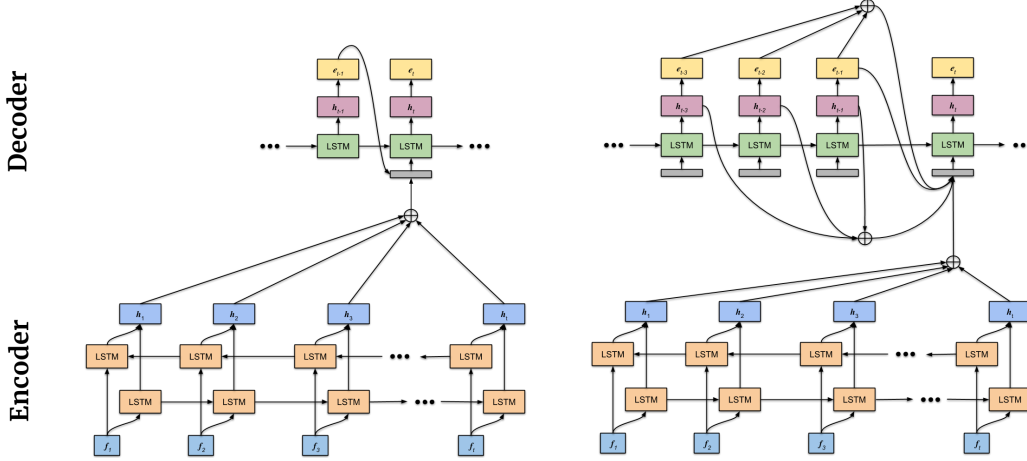

Figure 1: *Left:* Our baseline sequence-to-sequence model. *Right:* Our baseline model complemented with our dual self-attention mechanism. Note that the model visualized on the right, consists of two additional attention mechanisms: over the embedded outputs (yellow) and over the decoder's hidden states (light purple). Full sized diagrams may be found in the supplementary materials.

where $\mathbf{d}_t$ is defined as a weighted sum over the embedded outputs, $\{\mathbf{e}_1, \mathbf{e}_2, \ldots \mathbf{e}_{t-1}\}$. On the other hand, Daniluk et al. (2017) perform the attention over the decoder's hidden states rather than the outputs, resulting in the following recurrence function:

$$\mathbf{h}_t^d = S(\mathbf{h}_{t-1}^d, \mathbf{e}_{t-1}, \mathbf{c}_t, \widetilde{\mathbf{h}}_t) \tag{6}$$

where $\widetilde{\mathbf{h}}_t$ is a weighted sum over the decoder's past hidden states, $\{\mathbf{h}_1^d, \mathbf{h}_2^d, \ldots \mathbf{h}_{t-1}^d\}$. We propose to use *dual self-attention*, a novel combination of these two forms of self-attention, due to the complementary benefits that they provide. Our recurrence function therefore becomes:

$$\mathbf{h}_t^d = S(\mathbf{h}_{t-1}^d, \mathbf{e}_{t-1}, \mathbf{d}_t, \mathbf{c}_t, \widetilde{\mathbf{h}}_t) \tag{7}$$

where $\mathbf{d}_t$, $\mathbf{c}_t$ and $\widetilde{\mathbf{h}}_t$ are weighted sums of embedded past outputs, encoder hidden states and decoder hidden states, respectively.

The attention over the embedded outputs provides the decoder with direct information about the previously generated terms, allowing it to model non-sequential dependencies between output words (Werlen et al., 2018). On the other hand, the attention over the decoder's hidden states best deals with long-term dependencies, as it counteracts the problem of vanishing gradients (Bengio et al., 1994; Hochreiter, 1998) by allowing us to directly backpropagate time-steps other than the previous one.

### 3.3 Middle-Out Decoder

Unlike standard decoding, which typically begins from a START token and expands the sequence rightwards, middle-out decoding begins with an initial middle word and simultaneously expands the sequence in both directions.

The procedure by which we determine the value of the initial middle word is highly dependant on the application. For example, for action-focused video caption generation, a natural choice for the middle word would be a verb; for object-centric image captioning a better choice might be a prominent noun. In both cases, such words, which constitute a sub-set of the vocabulary, would need to be estimated from the encoded input itself. To accomplish this, we first utilize a classifier to predict the initial word and then use a middle-out decoder that begins with the initial word and expands the sequence outwards. Notably, in some scenarios a user may want to supply the initial word directly to the algorithm, resulting in a user-centric caption generation; this is something we explore in experiments.

**Middle Word Classifier:** The middle word classifier passes the input sequence through an LSTM encoder, to obtain the final hidden state $\mathbf{h}_t^e$. This hidden state is passed through a softmax layer, to obtain a probability distribution over the vocabulary. Note, other classifiers may also be used.

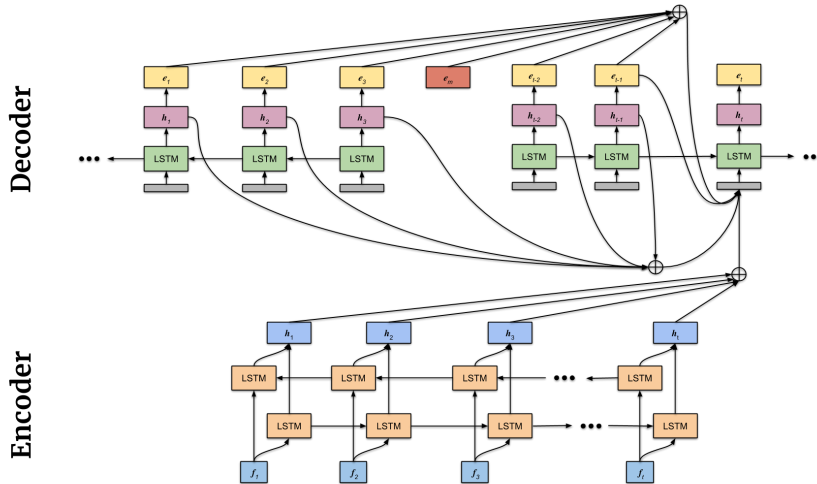

Figure 2: A diagram visualizing our middle-out decoder, complemented with our dual self-attention mechanism. Note that the two self-attention mechanisms combine the otherwise detached decoders. The middle-word shown in red is also attended over, allowing the two decoders to effectively depend on the output of the classifier (not pictured; can be found in the supplementary materials).

**Middle-Out Architecture:** The middle-out decoder consists of two LSTM decoders, one which decodes to the right and another which decodes to the left. They both begin from the same initial word, outputted by the middle word classifier, and the same initial hidden state, the final hidden state of the encoder. Both during training and inference, the middle-out decoder alternates between generating a word on the left and a word on the right, with both decoders continuing until they hit their respective STOP tokens.

In our vanilla middle-out model, there is no interaction between the left and right decoders and it is likely the case that the two decoders may generate sequences that are not jointly coherent. To combat this issue, we utilize self-attention as described in the next section.

**Middle-Out with Self-Attention:** To ensure coherence in the output of the middle-out decoder, we utilize the previously described dual self-attention as a communication mechanism between the left and right decoders. Our recurrence function, after the addition of the self-attention mechanism, is shown in Eq. (7). The model is visualized in Figure 2.

Our values for $\mathbf{d}_t$ and $\widetilde{\mathbf{h}}_t$ are computed over the outputs of both the left and the right decoder without any distinction, providing the two, otherwise disjoint, decoders the ability to communicate and ensure a logically consistent output. The value of $\mathbf{d}_t$, the weighted sum over all the embedded outputs, provides the decoders with information that wouldn't have otherwise been observed, namely the outputs of the other decoder. The value of $\widetilde{\mathbf{h}}_t$, the weighted sum over all the decoder hidden states, also accomplishes this in addition to providing each decoder with the ability to influence the gradients of the other decoder. As described earlier, self-attention allows a decoder to backpropagate to non-adjacent time-steps, however, in the case of the middle-out decoder, it also allows two disjoint decoders to backpropagate to entirely disconnected time-steps.

Effectively, self-attention allows the middle-out decoder to maintain some notion of dependence between all of the outputs, without necessitating the use of a single sequential decoder.

## 4   Experiments

We evaluate the aforementioned models on two sequence generation tasks. First, we evaluate middle-out decoders on the synthetic problem of de-noising a symmetric sequence. Next, we explore the problem of video captioning on the MSVD dataset (Chen and Dolan, 2011), evaluating our models for quality, diversity, and control.

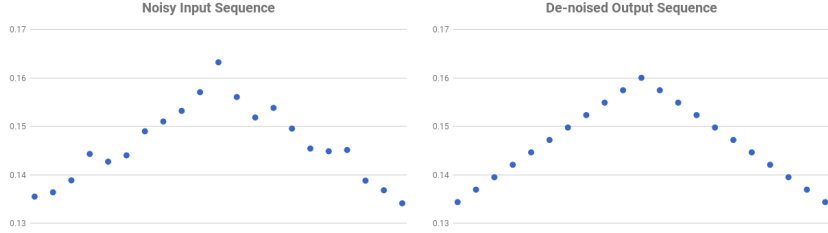

Figure 3: A visualization of the synthetic de-noising task. On the left, we have a noisy input sequence with each value being adjusted accordingly by an error sampled from a uniform distribution. On the right, we have the target output sequence, which has been de-noised.

## 4.1 Synthetic De-noising Task

**Dataset:** We first generate a symmetric 1-dimensional sequence of numbers, the output sequence, and proceed to add noise to it, forming the 1-dimensional input sequence. To generate our symmetric sequence of numbers, we first sample $\mu$ from $Uniform(-1, 1)$ to serve as the middle number in our sequence. Defining $\sigma = \mu^2$, we proceed to construct our sequence, $\mathbf{y}$, of length $2N + 1$ as follows:

$$\mathbf{y} = \mu - \frac{N\sigma}{N}, ..., \mu - \frac{2\sigma}{N}, \mu - \frac{\sigma}{N}, \mu, \mu - \frac{\sigma}{N}, \mu - \frac{2\sigma}{N}, ..., \mu - \frac{N\sigma}{N} \tag{8}$$

We generate $1000$ training samples and $100$ testing samples, with the value of $N$ being randomly selected such that $5 \leq N \leq 10$, resulting in sequences with lengths between $11$ and $21$.

To generate the noisy input sequence, we add noise sampled from $Uniform(-0.0035, 0.0035)$ to each element in the sequence ultimately obtaining our noisy input sequence, $\mathbf{x}$, such that $\mathbf{x}_i = \mathbf{y}_i + Uniform(-0.0035, 0.0035)$. An example of an input sequence and corresponding output sequence is shown in Figure 3.

**Experimental Setup:** We modify our models in order to best suit the nature of this task. Rather than treating this as a classification problem at each time-step, as is typically done for text generation, we instead formulate it as a regression. As a result of our model outputting a real-valued number, we remove the softmax layer, the embeddings layer, and the self-attention mechanism.

We opt to utilize a classifier to predict the initial value of the sequence for both the baseline model and the middle-out decoder. In both cases, the classifier is a single linear layer that operates on the final hidden state of a bidirectional LSTM encoder and ultimately outputs a single number. All of our modified models are visualized and further described in the supplementary materials.

We utilize 100-dimensional LSTMs and the Adam optimizer (Kingma and Ba, 2015) with a learning rate of $1e-4$. We train the models for $20,000$ steps with a batch size of $32$, which is equivalent to $640$ passes over the data. During training we use teacher forcing, meaning we feed the ground-truth output as input into the next time-step. During evaluation, we instead provide the model's own output as input into the next time-step.

**Results:** We evaluate the de-noising of the sequences with two metrics, MSE and symmetric MSE. Mean squared error (MSE) indicates the quality with which the model de-noised the sequence. Symmetric MSE measures the error between the two halves of the output sequence. Since the sequence is symmetric, theoretically the outputted sequence should have a very low error between its two halves. A higher error is indicative of a lack of coherence in the generation.

The results in Table 1 show that our middle-out decoder significantly improves upon the performance of the sequence-to-sequence baseline, with a $75\%$ decrease in the MSE value. There is an even larger gain in terms of the output symmetry, with an MSE value $570$ times smaller. This suggests that the middle-out decoder better models non-trivial dependencies in its outputs both through the use of its decoding order and self-attention mechanism.

| Models | MSE | Symmetric MSE |
|---|---|---|
| Sequence-to-Sequence with Attention | $1.52 \times 1\mathrm{e}{-3}$ | 0.0780 |
| Middle-Out with Self-Attention | $3.51 \times 1\mathrm{e}{-4}$ | $1.32 \times 1\mathrm{e}{-4}$ |

Table 1: Sequence generation results on a synthetic de-noising task. Middle-out decoding is shown to perform better on this task, both in terms of the quality and the symmetry of the de-noised sequence.

| Models | METEOR | CIDEr-D | ROUGE-L | BLEU-4 |
|---|---|---|---|---|
| Previous Work | | | | |
| LSTM-YT (V) (Venugopalan et al., 2015b) | 26.9 | - | - | 31.2 |
| S2VT (V + A) (Venugopalan et al., 2015a) | 29.8 | - | - | - |
| Temporal Attention (G + C) (Yao et al., 2015) | 29.6 | 51.7 | - | 41.9 |
| LSTM-E (V + C) (Pan et al., 2016b) | 31.0 | - | - | 45.3 |
| p-RNN (V + C) (Yu et al., 2016) | 32.6 | 65.8 | - | 49.9 |
| HNRE + Attention (G + C) (Pan et al., 2016a) | 33.9 | - | - | 46.7 |
| hLSTMat (R) (Song et al., 2017) | 33.6 | 73.8 | - | **53.0** |
| Our Models | | | | |
| Seq2Seq + Attention (I) | 34.0 | 78.9 | 70.0 | 47.4 |
| Seq2Seq + Attention + Self-Attention (I) | **34.4** | **81.9** | **70.5** | 48.3 |
| Middle-Out + Attention (I) | 30.9 | 68.6 | 66.9 | 40.8 |
| Middle-Out + Attention + Self-Attention (I) | 34.1 | 79.5 | 69.8 | 47.0 |
| Oracle Experiments | | | | |
| Seq2Seq + Attention (I) | 34.5 | 77.8 | 70.4 | 47.4 |
| Middle-Out + Attention + Self-Attention (I) | **40.9** | **124.4** | **78.6** | **62.5** |

Table 2: Video captioning results on the MSVD (Youtube2Text) dataset. We show previous work, our models as well as our oracle experiments. In this table, V, A, G, C, R, I respectively refer to visual features obtained from VGGNet, AlexNet, GoogLeNet, C3D, Resnet-152 and Inception-v4. It is worth noting that we only compare to single-model results. For reference, the state-of-the-art ensemble model (Pasunuru and Bansal, 2017) obtains METEOR: 36.0, CIDEr-D: 92.4, ROUGE: 92.4, BLEU-4: 54.5.

## 4.2 Video Captioning

Video captioning is the task of generating a natural language description of the content within a video. For this task, we utilize the MSVD (Youtube2Text) dataset (Chen and Dolan, 2011) to evaluate the output quality, diversity, and control of the models described in Section 3.

**Dataset:** The MSVD (Youtube2Text) dataset (Chen and Dolan, 2011) consists of 1970 YouTube videos with an average video length of 10 seconds and an average of 40 captions per video. We use the standard splits provided by Venugopalan et al. (2015a) with 1200 training videos, 100 for validation and 670 for testing. We utilize frame-level features provided by Pasunuru and Bansal (2017). The videos were sampled at $3 fps$ and passed through an Inception-v4 model (Szegedy et al., 2017), pretrained on ImageNet (Deng et al., 2009), to obtain 1536-dim feature vector for each frame.

**Experimental Setup:** For all of our models, we use a 1024-dimensional LSTMs, 512-dimensional embeddings pretrained with word2vec (Mikolov et al., 2013), and the Adam optimizer with a learning rate of $1\mathrm{e}{-4}$. We utilize a batch size of 32 and train for 15 epochs. We train our models with a cross entropy loss function. We employ a scheduled sampling training strategy (Bengio et al., 2015), which has greatly improved results in image captioning. We begin with a sampling rate of 0 and increase the sampling rate every epoch by 0.05, with a maximum sampling rate of 0.25.

During evaluation, we use beam search, with a beam size of 8 and a modified beam scoring function (Huang et al., 2017) which adds the length of the output sequence to the beam's score.

The middle-word classifier is trained to predict the most-common verb among all of the captions for a particular video. To avoid potential redundancies in our classifier's vocabulary (of size 40), we only predict words that end in *"ing"*. We utilize a cross-entropy loss function to train the model.

**Output Quality:** To evaluate quality of the captions produced by our models, we use four evaluation metrics popular for language generation tasks: METEOR (Denkowski and Lavie, 2014), BLEU-

| Classifier Accuracy | METEOR | CIDEr-D | ROUGE-L | BLEU-4 | Sampling Ratio |
|---|---|---|---|---|---|
| 31.64% Accuracy | 34.1 | 79.5 | 69.8 | 47.0 | 100% |
| 50% Accuracy | 35.6 | 90.4 | 71.9 | 50.3 | 73.14% |
| 75% Accuracy | 38.4 | 105.6 | 75.1 | 56.2 | 36.57% |
| 100% Accuracy | **40.9** | **124.4** | **78.6** | **62.5** | 0% |

Table 3: We simulate various classification accuracies by sampling from the output of our middle-word classifier and the oracle middle-words at different rates. We observe that we generate progressively better captions with improvements in classification accuracy.

4 (Papineni et al., 2002), CIDEr-D (Vedantam et al., 2015) and ROUGE-L (Lin, 2004). We use evaluation code from Microsoft COCO server (Chen et al., 2015).

The results shown in Table 2 demonstrate that the quality of the middle-out decoding is comparable to that of our sequence-to-sequence baseline. Our dual self-attention mechanism applied to the baseline outperforms previous work, and significantly improves the results of the middle-out decoder.

We note that the quality of our model depends on the quality of the middle word (verb) classifier. Our classifier for illustrative purposes is simple, but could be substantially improved using specialized action classification architectures, *e.g.*, dual-stream C3D networks (Carreira and Zisserman, 2017). To this end we perform an experiment to evaluate the performance of our model when provided with external input from an oracle. Rather than predicting the middle-word with a classifier, we instead provide the ground-truth to the model during inference. As our baseline sequence-to-sequence model is incapable of receiving an external input, we re-train a modified model in which the decoder receives the embedded ground-truth middle-word as input every time-step; *i.e.* both approaches use same data.

The results shown in Table 2, demonstrate that despite the fact that the baseline in our oracle experiment was specifically trained for the task at hand, our middle-out decoder performs significantly better, achieving a METEOR score of 40.9. This indicates that the middle-out decoder effectively utilizes external input to generate optimal captions.

In Table 3 we depict a series of experiments with various simulated accuracies for the middle-word classifier, by sampling from our classifier (accuracy of 31.64%) and the oracle middle-words at different ratios. This experiment demonstrates that reasonable improvements to the middle-word classifier would lead to strong performance gains in sequence generation.

**Output Diversity:** We quantify the diversity of the models by running a beam search, with a beam size of 8, and evaluating the diversity between the generated captions for each video. To evaluate diversity between the 8 captions, we employ Self-BLEU (Zhu et al., 2018) and Self-METEOR to assess similarity between the different captions in a generated set. The results in Table 4 demonstrate that the middle-out decoder significantly improves on the diversity of the generation caption set.

| Models | Self METEOR | Self BLEU-4 |
|---|---|---|
| Seq2Seq + Attention | 49.6 | 77.8 |
| Middle-Out + Attention + Self-Attention | **44.8** | **70.1** |

Table 4: Diversity evaluation (lower is better) our for a generated set of captions. We compare our baseline model as well as the middle-out decoder with self-attention.

**Output Control:** We define the output control to be a measure of the model's susceptibility to external input. A model which is able to effectively use external input to improve its generated caption can be said to be well controllable. The results of our oracle experiment, shown in Table 2, demonstrate our middle-out decoder to effectively utilize external input.

We expand on the oracle experiment by comparing to alternative mechanisms for controlled sequence generation, namely Grid Beam Search (Hokamp and Liu, 2017). The results shown in Table 5 demonstrate that our middle-out decoder is better at effectively leveraging the oracle information to improve the quality of the generated caption.

To further qualify the controllable nature of the middle-out decoder, we perform an experiment that exemplifies the decoder's ability to attend to varying parts of the video dependant on the input middle-word. We concatenate two videos together, without any explicit distinction about the boundary

| Models | METEOR | CIDEr-D | ROUGE-L | BLEU-4 |
|---|---|---|---|---|
| Seq2Seq + Attention (M) | 34.5 | 77.8 | 70.4 | 47.4 |
| Seq2Seq + Attention (G) | 40.4 | 121.6 | 77.7 | 61.0 |
| Seq2Seq + Attention (M) (G) | 40.0 | 120.0 | 77.8 | 60.5 |
| Middle-Out + Attention + Self-Attention | **40.9** | **124.4** | **78.6** | **62.5** |

Table 5: An expansion of the oracle experiment, where we evaluate the ability of each model to effectively utilize external input, in this case, a provided middle-word. Here (M) indicates that the model was trained with middle-word embedding concatenated to its input at each time step. (G) represents the usage of Grid Beam Search (Hokamp and Liu, 2017) during inference.

| | 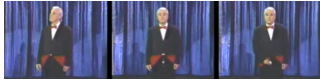 *a magician is performing magic* |  *a man is playing piano* |
|---|---|---|
| Provided Middle-Word | performing | playing |
| Seq2Seq + Attention | *a man is playing the piano* | *a man is playing the piano* |
| Middle-Out + Self-Attention | *a man is performing on a stage* | *a man is playing the piano* |

Table 6: The first row contains the sampled frames from the concatenated videos, as well as the ground-truth captions for each video. The row below indicates the ground-truth word provided to the model, as done in the oracle experiment. The proceeding rows are the outputs generated by our baseline model (specially trained for the oracle experiment) and our middle-out decoder, when given the specified input word. More qualitative examples are provided in the supplementary materials.

between the two videos, and demonstrate that by providing different initial middle-words to the middle-out decoder, we can effectively force it to generate captions pertaining to one of the videos.

The qualitative results of this experiment, shown in Table 6, demonstrate that the middle-out decoder is more susceptible to external input and therefore easier to control. Despite being trained to receive a ground-truth middle-word, the baseline model is incapable of effectively utilizing provided information, often choosing to ignore it entirely.

**Dual Self-Attention Ablation:** We perform an ablation study in order to demonstrate the effectiveness of our novel dual self-attention mechanism. The experiments shown in Table 7 demonstrate that dual self-attention outperforms the alternate methods for self-attention (Werlen et al., 2018; Daniluk et al., 2017), when utilized with our self-attention sequence-to-sequence architecture.

| Models | METEOR | CIDEr-D | ROUGE-L | BLEU-4 |
|---|---|---|---|---|
| Output Embedding Self-Attention | 34.2 | **82.8** | 70.3 | 47.8 |
| Hidden Self-Attention | **34.4** | 80.7 | **70.5** | 47.2 |
| Dual Self-Attention | **34.4** | 81.9 | **70.5** | **48.3** |

Table 7: An ablation study demonstrating the effectiveness of our novel dual self-attention mechanism.

# 5 Conclusion

In this paper, we present a novel middle-out decoder architecture which begins from an important token and expands the sequence in both directions. In order to ensure the coherence of the middle-out generation, we introduce a novel dual self-attention mechanism that allows us to model complex dependencies between the outputs. We illustrate the effectiveness of the proposed model both through quantitative and qualitative experimentation, where we demonstrate its capability to achieve state-of-the-art results while also producing diverse outputs and exhibiting controlability.

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
