[Reviews · NeurIPS 2018]

Reviewer 1



This paper describes a middle-out decoding scheme for the encoder-decoder frameworks, and its application to a synthetic symmetric task and the video captioning task. The main idea is that the decoder starts from an externally provided middle word (either an oracle, or a word predicted using a classifier over the vocabulary), and then alternates between expanding to the left and to the right. The two directions are connected through a self attention mechanism which uses two heads, one over the previous decoder hidden states, and one over the decoder inputs. The use of two attention heads for self attention is claimed as a novel contribution, but seems to be more of a design decision. The use of two heads instead of one is never tested directly. Strong results are shown on a synthetic task, and they are able to match the baseline on image captioning while increasing diversity. They can also easily control the output by changing the provided middle word. This is a clear paper with an interesting main idea and sound experimental design. I’m not sure the idea is sufficiently developed to warrant publication at this stage, but it is very close. I’m also not sure it’s a very good idea, which is affecting my grade somewhat, though I am trying my best to judge the work primarily based on its evaluation of this idea. At its core, middle-out decoding re-introduces some amount of pipelining into the encoder-decoder architecture. The system designer must decide how the middle word will be selected from the reference sentence (it isn’t the exact middle word by token count, but some high-information content word like a main verb or the subject or or object noun phrase), and they must decide how that middle word will be predicted before decoding begins. Those decisions will have a large impact on the ceiling for the complete system that will be built around them. I consider this pipeline to be a pretty steep cost. What we are buying with this cost is controllability and diversity. I do not see controllability as a large gain. Controllable output has been studied in NMT, and I’ll provide some references below. And the diversity effect isn’t compared against other methods of introducing diversity (again, I’ll provide references). So, I’m not sure if this cost is worth what we gain. Hence my middling score. There are a number of things the authors could do to make this a stronger paper: Implement the proposed future work and use a strong, proven action classification architecture to predict the middle word, and beat the baseline of Seq2Seq + Attention + Self-Attention. I suspect that if this had happened, I would not be focusing so much on the side-benefits. Another way to potentially improve your score would be to try incorporating the middle word choice into the beam, so with a beam of 5, the system starts with the 5-best choices for the middle word from the current model. That is, if you are not already doing so (it isn’t clear from the paper). Compare to known methods for controlling the words in the output sequence by altering the beam search: https://arxiv.org/abs/1804.06609 https://arxiv.org/abs/1704.07138 I suspect you will win here, as you are altering both the search and the model, while the above techniques alter only the search, but without experiments it’s hard to know. Providing both the embedding of the constraint-word to the decoder while also constraining the search to output the word would be an interesting and novel way to extend the work cited above to change both the model and search. Compare to a known methods for introducing diversity, for example (though you can probably do better than these, there may well be techniques specific to the image captioning domain that I am unaware of): https://arxiv.org/abs/1611.08562 https://arxiv.org/abs/1605.03835 Other, smaller concerns: Equation (3) has a circular dependency on h^d_t, you probably meant h^d_{t-1}. Do you share encoders for the middle word classifier and the middle-out decoder? If so, please specify how you balance the two loss functions. If not, it might be worth considering to strength the middle-word classifier. Your results on the synthetic data are impressive, but it would be worthwhile to specify how many more parameters are used with this new model. Likewise for the captioning experiments. === Thanks very much for addressing my comments in your response. Given these additional experiments and comments, I will gladly raise my scores. Using a sampled oracle to simulate a higher accuracy classifier was particularly clever. Good luck with your future research. ===

Reviewer 2



The paper presents a novel decoding strategy for seq2seq models, which starts from the middle of the output and goes both left and right. To make the left and right branches consistent the authors suggest to add attention over both left and right states and outputs. The demonstrate their approach on a synthetic task and on a youtube (YT) to captions task and show promising results. On the YT task they use a simple predictor to decide on the first word and show that there result is competitive with a strong baseline. Moreover - they show that if the first word is known (oracle) they get nice gains over a suitable baseline. The method is motivated by 1. Need to enforce a particular word in the middle of the output. 2. Dealing with (somewhat) symmetric sequences. The paper is clearly written. The approach is novel, the formulation is sound and the experiments are convincing. This is a natural tool and I believe it is of interest for the seq2seq compunity.

Reviewer 3



The paper improves the decoder of an attention-based seq2seq model in two ways: 1. Middle-out decoding: the left-to-right RNN decoder is replaced with 2 RNNs, one going left-to-right and the other going right-to-left, both generating a half of the sentence and both conditioned on an important middle word This allows easier control of the middle word, a task otherwise hard for RNNs. It also needs additional supervision in indicating the important middle word. 2. The decoders run the attention mechanism also over past outputs, and past hidden states. This is especially important for the middle-out case as it is its only way of synchronizing the two decoders. The main benefit of the middle-out decoding is enhanced ability of controlling the model to output a given word - the network is literally forced to do so through conditioning on the middle word. The enhanced self-attention is required by the middle-out decoder to be self-consistent, as it allows the two RNNs to see each other's outputs and hidden states (another obvious way of doing this would be to fully run one of he the RNNs, e.g. generating the first half of the sentence, then condition the other RNN on the output of the first). The paper doesn't specify what is the sequence of evaluating the left-to-right and right-to-left rnns (I assume they alternate, essentially generating the sequence in order n/2, n/2+1, n/2-1, n/2+2, n/2-1,.../n,1), but this should be clarified in paragraph l.161-175. On a toy task (denoising a symmetrical sequence corrupted with elementwise uniform noise) the middle-out decoding outperforms the unidirectional RNN, however it has the benefit of implicitly knowing the important middle input (it it separately trained to predict it. I am curious if the middle-out decoder would also be more efficient if the location of the peak was also randomized. On a real-world task the results indicate that: 1. The vanilla seq2seq model slightly improves with the addition of self-attention 2. The middle-out decoder slightly underperforms the vanilla RNN, both with and without self-attention. It also clearly needs the self-attention. 3. On a dataset containing concatenations of videos, the middle-out decoder correctly selects the video to caption based on the most important word provided to it, while the seq2seq model often ignores the conditioning word. However, for a fair baseline one could constrain the beam search to emit the provided word by e.g. running several beam searches, each forcing the emission of the word at a certain location - the grammar of the captions is simple enough that this strategy should work. I am unsure about the fairness of comparing sample diversity using beam search, as it is a tool to find the most likely output and by design concentrates around the mode of a probability distribution. To get diverse samples, one should sample from a model. If the network is certain about the proper output the variations in the beam will concentrate near the end of the sequence. The middle-out decoder effectively has two ends (it does two beam searches in parallel), so it may naturally have more diversity - it has variations in the beginning and end of a sequence, rather than just at the end. However, even an unconditional language model, when searched with a beam search, will generate a fairly consistent set of hypotheses and for diversity it should be sampled from. The novelty of the paper is fairly low: - the self-attention has been shown to improve language models in many prior papers, it is also older than the references given in the paper (e.g. https://arxiv.org/abs/1601.06733) - the solution to run a forward and backward LM to force the emission even if not published before is an obvious baseline to try Minor remarks: - the order of evaluation of the left-to-right and right-to-left RNNs should be specified in the text - The experiments in Table 3 and 4 slightly confuse the impact of the middle-out decoding and self-attention - why is the vanilla decoder tested without it and the middle-out decoder tested with it? - earlier use of self-attention for text modeling is https://arxiv.org/abs/1601.06733 and it should be included in the references